# Effect of Neoadjuvant Therapies on Soft Tissue Sarcomas with Tail-like Lesions: A Multicenter Retrospective Study

**DOI:** 10.3390/cancers13153901

**Published:** 2021-08-02

**Authors:** Hisaki Aiba, Kunihiro Ikuta, Kunihiro Asanuma, Katsuhisa Kawanami, Satoshi Tsukushi, Akihiko Matsumine, Daisuke Ishimura, Akihito Nagano, Yoji Shido, Eiji Kozawa, Kenji Yamada, Junji Wasa, Hiroaki Kimura, Takao Sakai, Hideki Murakami, Tomohisa Sakai, Tomoki Nakamura, Yoshihiro Nishida

**Affiliations:** 1Department of Orthopedic Surgery, Graduate School of Medical Sciences, Nagoya City University, Nagoya 467-8601, Japan; hikimura@med.nagoya-cu.ac.jp (H.K.); sts-orth@med.nagoya-cu.ac.jp (T.S.); hmuraka@med.nagoya-cu.ac.jp (H.M.); 2Department of Orthopaedic Surgery, Nagoya University Graduate School of Medicine, Nagoya 466-8560, Japan; k-ikuta@med.nagoya-u.ac.jp (K.I.); tosakai@med.nagoya-u.ac.jp (T.S.); ynishida@med.nagoya-u.ac.jp (Y.N.); 3Department of Orthopaedic Surgery, Mie University Graduate School of Medicine, Tsu 514-8570, Japan; kasanum@med.mie-u.ac.jp (K.A.); tomoki66@med.mie-u.ac.jp (T.N.); 4Department of Orthopaedic Surgery, Aichi Medical University School of Medicine, Nagakute 480-1195, Japan; kawanami.katsuhisa.904@mail.aichi-med-u.ac.jp; 5Division of Orthopaedic Surgery, Aichi Cancer Center Hospital, Nagoya 464-8681, Japan; s-tsuku@aichi-cc.jp; 6Department of Orthopaedics and Rehabilitation Medicine, Faculty of Medical Sciences University of Fukui, Fukui 910-1193, Japan; matsumin@u-fukui.ac.jp; 7Department of Orthopaedic Surgery, Fujita Medical University, Toyoake 470-1192, Japan; isssi-05@fujita-hu.ac.jp; 8Department of Orthopaedic Surgery, Gifu University Graduate School of Medicine, Gifu 501-1194, Japan; a-nagano@lucky.odn.ne.jp; 9Department of Orthopaedic Surgery, Hamamatsu Medical University, Hamamatsu 431-3125, Japan; shido@hama-med.ac.jp; 10Department of Orthopaedic Surgery, Nagoya Memorial Hospital, Nagoya 468-8250, Japan; e.kozawa.h@hospy.or.jp; 11Department of Orthopedic Oncology, Okazaki City Hospital, Okazaki 444-8553, Japan; yamada.kenji@okazakihospital.jp; 12Division of Orthopaedic Oncology, Shizuoka Cancer Center Hospital, Nagaizumi 411-0934, Japan; j.wasa@scchr.jp; 13Department of Rehabilitation, Nagoya University Hospital, Nagoya 466-8560, Japan

**Keywords:** soft tissue sarcoma, invasive front, tail-like lesion, myxofibrosarcoma, undifferentiated pleomorphic sarcoma, neoadjuvant therapy, radiotherapy, chemotherapy

## Abstract

**Simple Summary:**

It is essential to focus on the tumor invasive front (tail-like lesion)—the soft tissue sarcoma’s specific peripheral infiltrative growth characteristics—to avoid leaving unexpected tumor residues during surgery. This study aimed to analyze the effect of neoadjuvant therapy for highly malignant soft tissue tumors with tail-like lesions. From 2012 to 2019, 36 patients were treated with neoadjuvant therapy, including chemotherapy, radiotherapy, or both. Consequently, we observed shrinkage, and occasionally the disappearance of the tail-like lesion. The lesion’s regression was related to the necrosis rate of the main part of the tumor. However, the regression of lesions was not directly related to the achievement of surgery with a microscopically negative margin or improvements of oncological outcomes. Thus, a more multi-angle evaluation to elaborate surgical strategy is necessary.

**Abstract:**

Several types of soft tissue sarcomas have peripheral infiltrative growth characteristics called tail-like lesions. The efficacy of neoadjuvant therapy for tumors with tail-like lesions has not been elucidated. From 2012 to 2019, we analyzed 36 patients with soft tissue sarcoma with tail-like lesions treated with neoadjuvant therapy, including chemotherapy, radiotherapy, or both. The effect of neoadjuvant therapy on the tail sign was investigated by analyzing the change in tail-like lesions during neoadjuvant therapy and histological responses. The median length of the tail-like lesion reduced from 29.5 mm at initiation to 19.5 mm after neoadjuvant therapy. The extent of shrinkage in tail-like lesions was related to the histopathological responses in the main part of the tumor. Complete disappearance of the tail-like lesion was observed in 12 patients; however, it was not related to achieving a microscopically negative margin. The oncologic outcomes did not significantly differ between cases with and without the complete disappearance of tail-like lesions. This study indicated that the shrinkage of tail-like lesions did not have a significant effect on complete resection or improvements of clinical outcomes. A more comprehensive evaluation is needed to elaborate on the surgical strategy.

## 1. Introduction

Soft tissue sarcomas are rare and heterogeneous entities with local or distant metastatic potential [1]. Approximately 10–30% of patients experience local recurrences, complicating subsequent procedures and occasionally resulting in amputation [2]. Several soft tissue sarcoma types have peripheral infiltrative growth characteristics around the invasive fronts (tail-like lesions) [3,4]. The surgical intervention plan should include these reactive zones during complete resection [5,6].

Neoadjuvant therapy using radiotherapy, chemotherapy, or both is now considered, especially for locally advanced tumors, to improve resectability with appropriate margins and long-term oncologic outcomes. The National Comprehensive Cancer Network (NCCN) guideline, 2021 (NCCN Clinical Practice Guidelines in Oncology, Soft Tissue Sarcoma Version 2.2021—28 April 2021, https://www.nccn.org/professionals/physician_gls/pdf/sarcoma.pdf), recommends neoadjuvant therapies for resectable stage II–III patients with adverse functional outcomes. These methods include radiotherapy [7], chemoradiotherapy [8], or chemotherapy [9,10]. Due to heterogeneity, the contribution of chemotherapy to the improvement of oncologic outcomes of soft tissue sarcomas was considered to be limited [11]. However, if limited to high-risk cases (high-grade malignancy, ≥5 cm in diameter, and deeply located with respect to investing fascia), the efficacy of the chemotherapy has been indicated [12,13].

Thus far, little is known about the effects of these methods on tail-like lesions. This study aimed to analyze the effect of neoadjuvant therapy on tail-like peripheral lesions based on MRI and the histological evaluation of resected specimens.

## 2. Materials and Methods

### 2.1. Patients

We included patients with histologically diagnosed malignant soft tissue tumors with tail-like lesions who underwent neoadjuvant therapy for primary soft tissue tumors between January 2012 and December 2019. Certified pathologists confirmed all diagnoses at each hospital. We excluded patients with visceral location, metastasis (distant, skip lesion from the primary site, or lymph node metastasis) at diagnosis, or a lack of images for proper evaluation. Furthermore, we excluded patients who underwent amputation. Using independent questionnaires assigned to the 12 hospitals of the Tokai Musculoskeletal Oncology Consortium, 105 patients who underwent neoadjuvant therapy were extracted from a total of 951 patients. Among those, 36 patients who exhibited tail-like lesions were finally included in the study. The questionnaire included sex, age at diagnosis, histological diagnosis, histological grade according to the French Federation of Cancer Centers [14], tumor location and depth, and (neo)adjuvant therapy details.

Surgical margins were classified as microscopically negative (R0), macroscopically negative but microscopically positive (R1), and macroscopically positive (R2) [15]. Certified pathologists determined the categorized margin status with a review of the edge of the tumor and the extension of tail-like lesions. In addition, we collected information about skin reconstruction (graft or flap), prosthesis usage (e.g., total knee arthroplasty), and the necessity of manipulating major neurovascular bundles (AVN).

The study was approved by the Ethics Committee of Nagoya City University Hospital. The study design and procedures were conducted in accordance with the principles of the Declaration of Helsinki.

### 2.2. Neoadjuvant Therapy

In this study, neoadjuvant therapy was performed according to doctors’ preferred methods. For radiotherapy, the clinical target volume was expected to be the gross target volume, enhanced with a gadolinium T1-weighted image, plus tail-like lesion with 1–2 cm margin. Neoadjuvant external beam radiation was administered at 45–54 Gy/22–25 fr with permission for adjuvant radiation up to 60 Gy [16].

Chemotherapy was performed based on the standard chemotherapy in Japan: Adriamycin, 60 mg/m^2^ plus ifosfamide 10 g/m^2^ (AI) or gemcitabine 1800 mg/m^2^ plus docetaxel 70 mg/m^2^ (GD) in 3-week intervals [17,18]. In some institutions, etoposide was added to the AI regimen [19].

Moreover, chemoradiotherapy with hyperthermia was performed to augment the efficacy of chemoradiotherapy [20]. In this protocol, radiotherapy was administered to the primary site for a total of 40 Gy/20 fr. For thermotherapy, an 8 MHz radiofrequency capacitive heating system (Thermotron RF-8: Yamamoto VINITA, Osaka, Japan) was used for weekly hyperthermia with simultaneous chemotherapy [21,22].

### 2.3. Evaluation of Tail-Like Lesions and Response to Neoadjuvant Therapies

Tail-like lesions were evaluated using T1-weighted images with gadolinium enhancement or short TI-inversion recovery (STIR) images. A tail-like lesion was defined as “a curvilinear shaped tapered thick fascial enhancement extending from the primary mass, with or without irregularity of the tumor border.” The tail sign’s length was calculated from the base of the tail sign on the main mass to the top of the tail sign in the largest cross-sectional plane. The tail sign’s thickness was calculated as the length between the tail sign’s edges [23].

If *a* and *b* indicate the length/thickness of the tail-like lesion before and after neoadjuvant therapy, respectively, the change in the tail-like lesion was evaluated using the following formula:(1)$$\frac{a-b}{a}\times 100\;\left(\%\right)$$

The representative figures are shown in Appendix A.

Response evaluation criteria in solid tumors (RECIST) 1.1 were used to evaluate the main tumor’s response to neoadjuvant therapy. At each institution, a certified radiologist independently measured the greatest longitudinal dimension. The four response categories included in RECIST 1.1 are:Complete response (CR): disappearance of all target lesions;Partial response (PR): target lesion’s diameter decreases by >30%;Progressive disease (PD): target lesion’s diameter increases by >20%;Stable disease (SD): lesion that does not meet the other criteria.

The change in size was evaluated at the beginning of neoadjuvant therapy and immediately before surgery [24].

### 2.4. Histological Response

The four-tier histological response was defined as follows [18]:Grade 1: little or no effect of neoadjuvant therapy observed;Grade 2: partial response to neoadjuvant therapy with >50% tumor necrosis;Grade 3: >90% tumor necrosis attributable to preoperative neoadjuvant therapy, although foci of apparently viable tumors may be seen in some histologic sections;Grade 4: no apparent viable tumor cells observed in any histological section.

Certified pathologists at each hospital evaluated these histological responses.

### 2.5. Statistical Analysis

This study’s primary goal was to analyze the relationship between changes in tail-like lesions during neoadjuvant therapy and histological responses. The secondary goal was to analyze the effects of the pictorial changes (tail-like lesions or the main part (RECIST 1.1)) and four-tier histological responses. Paired *t*-tests and Mann–Whitney U tests were performed to compare the mean and histological responses before and after therapy. The correlations between the variables were evaluated using Pearson’s moment correlation coefficient (<0.2, no correlation; 0.2–0.4, weak correlation; 0.4–0.7, moderate correlation; and >0.7, strong correlation).

We also used the Kaplan–Meier method to estimate the overall survival (the time from diagnosis to death due to any cause), distant metastasis-free survival (the time from diagnosis to distant metastasis), and local relapse-free survival (the time from surgery to local recurrence). In the univariate analysis of oncologic outcomes, the differences between curves were analyzed using log-rank analysis. Potential risk factors for oncologic outcomes were analyzed using a stepwise Cox proportional hazards model, and hazard ratios (HRs) were calculated. We performed this accessory analysis to show the efficacy of neoadjuvant therapy by comparing the histologically and chronologically matched patients from the administrative hospital (Nagoya City University).

All statistical analyses were performed using SPSS Statistics for Windows, version 25 (IBM Corp., Armonk, NY, USA). Statistical significance was set at *p* < 0.05.

## 3. Results

### 3.1. Patient Characteristics

The study included 36 patients (21 males and 15 females; mean age at diagnosis, 57.9 ± 15.5 years), and the median follow-up period was 1362 days from the first visit (interquartile range (IQR), 1001–2000) and 1267 days after surgery (IQR, 886–1893). Histologically, the tumors were classified as undifferentiated pleomorphic sarcoma (UPS, *n* = 11), myxofibrosarcoma (MFS, *n* = 13), synovial sarcoma (SS, *n* = 4), dedifferentiated liposarcoma (DDL, *n* = 4), and others (one patient each with epithelioid sarcoma, malignant peripheral nerve sheath tumor, clear cell sarcoma, and extraskeletal Ewing sarcoma). The average tumor length was 76.0 mm (IQR, 53.3–93.3), the tail-like lesion’s length was 29.5 mm (IQR, 23.0–37.3) and thickness was 4.0 mm (IQR, 2.2–7.7) (Table 1).

The depth of the main lesion was evaluated as superficial (*n* = 13) or deep (*n* = 23). There were differences in the median length of tumors between the tumors in deep locations and superficial locations (85.0 mm (IQR, 65.0–112.0) and 57.0 mm (IQR, 45.0–80.0), *p* = 0.043, Mann–Whitney U tests, deep andsuperficial, respectively). There were no differences in the median length or thickness of tail-like lesions between the tumors in a deep location or superficial location (length, 30.0 mm [IQR, 20.0–44.0] or 28.0 mm (IQR, 24.0–32.0), *p* = 0.515, thickness, 4 mm (IQR, 2.3–7.8) or 3 mm (IQR, 2.2–5.0), Mann–Whitney U tests, deep or superficial, respectively).

### 3.2. Effectiveness of Neoadjuvant Therapy on the Main Mass or Tail-Like Lesion

The median maximum tumor length was 76.0 mm (IQR, 53.3–93.3) at initiation and 63.0 mm (IQR, 43.3–93.8) after neoadjuvant therapy, with no significant change during the treatment (*p* = 0.088). The mean rate of change in the maximum length was −5.0% ± 28.4% (median, −5.3%; IQR, −24.4% to + 12.6%) (Figure 1a). On the RECIST 1.1, patients were categorized as SD (*n* = 22 (61.1%), not confirmed), PR (*n* = 8 (22.2%)), or PD (*n* = 6 (16.7%)). Univariate and multivariate analyses for the achievement of PR are shown in Appendix A. There were no incisive biomarkers for anticipating a good response.

The median length of the tail-like lesion was 29.5 mm (IQR, 23.0–37.3), and the median thickness was 4.0 mm (IQR, 2.2–7.7) at initiation and 19.5 mm (IQR, 0–36.5; length) and 2.4 mm (IQR, 0–3.8; thickness) after neoadjuvant therapy. The mean rates of change of the tail-like lesion were −38.0% (±56.8%; median, −34.6% (IQR, −100 to −3.6)). In addition, the rate of change of the tail-like lesion’s thickness was −41.3% (±48.7%; median, −26.3% (IQR, −100 to 0) (Figure 1b,c). There was a statistically significant reduction in thickness (*p* < 0.001), but not in length (*p* = 0.088). Complete disappearance of the tail lesion was observed in 12 patients. The univariate and multivariate analyses for the achievement of complete disappearance are shown in Appendix A. There were no incisive biomarkers for anticipating a good response.

There was a weak or moderate relationship between the shrinkage of the maximum length of the tumor and the tail-like lesion in the Pearson’s moment correlation coefficient (length, r = 0.36, *p* < 0.001; thickness, r = 0.42, *p* = 0.047; Figure 1d,e). There was a strong relationship between the length and thickness of the tail-like lesions (r = 0.87, *p* < 0.001; Figure 1f).

In addition, the effectiveness of neoadjuvant therapy was depicted using waterfall plots for the maximum length of the tumor (main part) or tail sign (Figure 2a–c).

### 3.3. Histopathological Evaluation of the Resected Tumor

Histopathological efficacy was evaluated using a four-tier grading system. The responses were as follows: G1, 15 patients (42%); G2, 13 patients (36%); G3, seven patients (19%); and G4, one patient (3%) (Table 2).

The pie charts indicate the relationships between the neoadjuvant modalities and histological response (Figure 2d).

The median change in the maximum tumor length was +8.9% (IQR, −4.27 to 33.3) in G1 response, −12.1% (IQR, −27.8 to 7.9) in G2, and −33.3% (IQR, −43.5 to −2.2) in G3 + 4 patients. There were significant differences between G1 and G2 (*p* = 0.006), G1, and G3 + 4 (*p* = 0.001, Figure 3a). In addition, the median change in the length of the tail-like lesion was −8.5% (IQR, −100.0 to 0.0) in the G1 response, −45.2% (IQR, −100.0 to −14.9) in the G2 response, and −80.0% (IQR, −100.0 to −20.1) in G3 + 4 patients (Figure 3b). There were significant differences between G1 and G3 + 4 groups (*p* = 0.05). Moreover, the median change in the thickness of the tail-like lesion was –9.1% (IQR, –100.0 to 0.0) in the G1 response, –40.0% (IQR, –100.0 to 0.0) in the G2 response, and –89.3% (IQR, –100.0 to –16.2) in G3 + 4 patients (Figure 3c). There were significant differences between G1 and G3 + 4 groups (*p* = 0.03).

### 3.4. Impact of Neoadjuvant Therapy on Margin Status

Overall, 27 patients underwent R0 resection. The relationships between the margin status and patient characteristics, surgical procedure, and response to neoadjuvant therapy are summarized in Appendix A. The patients with superficial lesions underwent plastic surgery more frequently than those with deep lesions (the number of plastic surgeries = 4/23 or 10/13, deep or superficial, respectively, *p* < 0.001, chi-squared analysis). Univariate analysis revealed that patients other than UPS or MFS chemotherapy more frequently achieved R0 resection (the number of R0 resections = 15/24, 12/12, UPS + MFS, SS + DLS + others, respectively, *p* = 0.016, chi-squared analysis). Although there was no statistical difference in the patients with good responses to neoadjuvant therapy according to RECIST 1.1, there was no R1 resection in good responders (the number of R0 resections = 19/28, 8/8, SD or PD, PR, respectively, *p* = 0.06, chi-squared analysis). The disappearance of the tail-like lesion was not related to R0 resection (number of R0 resections = 9/12 or 18/24, patients with or without disappearance of the tail-like lesion, respectively, *p* = 1.0, chi-squared analysis). Moreover, there were no apparent differences in patients who underwent a difficult surgery, including skin reconstruction, manipulation of AVN, or insertion of a prosthesis.

### 3.5. Oncologic Outcomes of Neoadjuvant Therapy

We evaluated oncologic outcomes as accessory endpoints. Overall, regarding the oncologic outcome at the end of follow-up, seven patients died of disease, one died of another disease, four were alive with disease, and 24 had no evidence of disease. The overall survival was 85.1% ± 6.2%, distant relapse-free survival (D-RFS) was 65.5% ± 8.1%, and local relapse-free survival (L-RFS) was 92.6% ± 5.0% at five years. Univariate analysis results of these outcomes are summarized in Appendix A.

In addition, to determine the importance of neoadjuvant therapy, we compared patients who had no adjuvant therapy only in the UPS + MFS population. A total of 24 patients were compared to those who underwent neoadjuvant therapy (*n* = 24). Among them, three patients received adjuvant radiotherapy due to R1 resection. Kaplan–Meier curves for the groups with and without neoadjuvant therapy revealed that the overall survival was 82.4% ± 8.1% and 84.6% ± 8.3%, D-RFS was 58.8% ± 11.1% and 53.1% ± 13.6%, and L-RFS was 86.3% ± 9.2% and 65.8% ± 10.7%, respectively (Figure 4). Univariate analysis revealed that L-RFS was significantly higher in patients who received neoadjuvant therapy (*p* = 0.031; hazard ratio, HR = 0.21). There were no differences in overall survival (*p* = 0.64, HR = 1.34) and D-RFS (*p* = 0.93, HR = 1.04) (Figure 4). The differences in the basic characteristics are summarized in Appendix A. Neoadjuvant therapy was performed exclusively for younger patients, higher-grade tumors, lower extremities, and longer tail-like lesions.

## 4. Discussion

The tail-like sign was first introduced by Fanburg-Smith et al., in 1999 [25,26]. Tumor infiltration was pathologically proven in 83% of superficial malignant fibrous histiocytomas. The infiltrative growth pattern, connecting the tumor to the fascial plane and skeletal muscle without a discrete nodular lesion [27], is considered a primary risk factor for local recurrence [3,28].

This study partially focused on low-grade MFS, a myxoid variant of malignant fibrous histiocytoma [29]. Despite the low-grade characteristics of most lesions, the tumor has relentless recurrence potential [30], with a 40–60% recurrence rate [29,31]. Moreover, recurrence may transform the tumor to a higher grade [30]. This phenomenon makes it more challenging to treat recurrent tumors requiring multiple surgeries. Thus, a well-planned surgery using appropriate neoadjuvant therapy and the complete removal of possible extensions of the tumor is important in the primary setting.

The characteristics of tail-like lesions have been extensively discussed. In some cases, the lesion mainly consisted of reactive edema with no viable or invading tumor [5,32]. In this study, we could not prove the importance of the complete disappearance of tail-like lesions after neoadjuvant therapy, and the disappearance was not related to achieving R0 resection or improvement of oncological outcomes. This is partially because the complete disappearance of tail-like lesions consisting of edema and inflammation is not true regression of a tumor. An accurate image diagnosis to distinguish between actual and false tail-like lesions is necessary for a tumor’s ideal resection with adequate surgical margins to minimize damage to the adjacent important structures and maximize resectability without any residual tumor.

Histopathologically, the tail-like lesion comprised the viable tumor and infiltrated into the fascia or subcutaneous fat layer accompanied by fibrous tissue [26]. These viable tumors changed into necrotic tissue after effective neoadjuvant therapy. However, the tail-like lesion’s traces remained as empty fibrous tissue budding around the tumor. Therefore, it is difficult to distinguish whether the skin contains neoplastic cells. Histopathological analysis of 18 patients by Imanishi et al., reported that after preoperative radiotherapy, the tail sign contained a viable tumor in seven cases and a non-viable tumor in five cases. Likewise, we evaluated the actual effect of neoadjuvant therapy in tail-like lesions and proved the relationship between histological responses in the main tumor lesion and regression of the tail-like lesion. These findings indicate that neoadjuvant therapy’s efficacy in the main part can be a useful surrogate marker of efficacy in tail-like lesions.

We also showed that the achievement of R0 resection was related to the tumor subtype with high residue rates in UPS or MFS. Although not statistically significant, the patients who responded to the neoadjuvant therapy tended to achieve R0 resection, suggesting that effective neoadjuvant therapy and reactivity to therapy are essential for a tumor’s complete resection. However, we should take into consideration that even for a certified pathologist, it is difficult to evaluate the true extension of a tumor along with tail-like lesion after neoadjuvant therapy, which comprise fibrous tissue, fibroblast cells, or degenerated tumor. This implies that some cases of pathological evaluations of margin status might not be precise.

The effect of neoadjuvant therapy on the tail-like lesion remains controversial. Several studies have concluded that preoperative radiotherapy has no effect on the tail sign [28], although others have reported positive results [33,34]. These conflicting viewpoints were due to differences in the sample sizes of the studies, or the methods used to evaluate the efficacy of neoadjuvant therapy on tail-like lesions. Our provisional data suggested that neoadjuvant therapy improved the local control rate by comparing the histologically and chronologically matched patient cohorts. However, selection bias may have affected the results; therefore, a validation study is needed to confirm our findings by analyzing the prospective or data-matched cohorts.

Despite no statistical backing, we showed favorable results for the shrinkage of tail lesions and in histopathological necrosis grades in patients treated with chemoradiotherapy. In case of resistance to radiotherapy or chemotherapy by the soft tissue sarcoma, these multimodal agents might be considered. In addition to chemoradiotherapy, some institutions perform hyperthermia to augment the efficacy of chemotherapeutic agents; a recent phase-III randomized study (EORTC 62961) showed that regional hyperthermia increases the benefit of preoperative chemotherapy in patients with localized high-risk STS on comparing etoposide, ifosfamide, and doxorubicin (EIA) alone, with combined EIA and hyperthermia [20]. However, according to the NCCN 2021 guidelines, hyperthermia with preoperative chemotherapy is not recommended, and the results need to be confirmed in large cohort studies. The addition of hyperthermia influenced detection of the tail-like lesions, because the procedure induced inflammation around the target area.

This study has several limitations. First, our MRI evaluation detected the presence or disappearance of the tail sign in both contrast T1-weighted and STIR images. Theoretically, the former indicated tumor viability and the latter an edematous tissue [35]; therefore, detection bias should be considered. Secondly, several definitions of the tail-like sign have been proposed. Fanbeug-Smith defined a tail-like sign as “a pathological tumor extension along normal tissue planes for >2 mm from the edge of the main mass [3].” The pictorial definition by Ferenbro et al. modified the perspective as “an irregular surface with spicula-like extensions into the surrounding tissue of >25% of the circumference on an MR T2-weighted image.” Subsequent definitions described it as “a crawling change beyond the fascia [28,32],” “a well-defined, sharp or tapering, pointed curvilinear projection at least 1.0 cm in length on T1-weighted image with contrast [36],” and “a tapered fascial enhancement extending from the tumor margin with >2 mm thickness.” We used Yoo et al.’s definition [23]; a different definition might affect the reproducibility of this study. Thirdly, our multicenter study permitted various procedures as “neoadjuvant therapy” because the study’s primary objective was to analyze the changes in tail-like lesions during neoadjuvant therapy, but not the oncological outcomes. Thus, the chemotherapy’s intensity or the area of radiotherapy should be normalized while focusing on the oncological outcomes. However, in some institutions in Japan, the physicians do not perform routine radiotherapy because the Japanese Orthopaedic Association clinical practice guidelines on the management of soft tissue tumor do not recommend routine radiotherapy [37], despite decent evidence in support of radiotherapy [7,38]. Some clinicians in Japan prefer to perform perioperative chemotherapy based on the decent result of clinical studies (JCOG0304 [18], JCOG1306 [17]) performed in our nation. Differences among the modalities should be validated in a future study. Finally, the evaluation of the response to neoadjuvant therapy, including radiotherapy, is difficult based only on the size of the tumor. Specifically, Canters et al. reported a steady size after neoadjuvant radiotherapy despite some patients having responded decently to radiotherapy [39]. Although RECIST 1.1 is ubiquitously used as the tool for the evaluation of the response to some agents, this criterion is based only on the size. We should consider the other criteria, including the combination of accumulation of contrast agents or radioisotopes in a future study.

Nonetheless, this study is the first to analyze the effect of neoadjuvant therapy on soft tissue tumors with tail-like lesions. Further research is expected to validate the data in a more sophisticated manner.

## 5. Conclusions

Our multicenter study analyzed the effect of neoadjuvant therapy on the tumor invasive front or ‘tail-like lesion.’ After neoadjuvant therapy, tail-like lesion shrinkage was observed in many patients and was related to the effect on the main part of the tumor; however, we could not confirm the relationship between shrinkage of tail-like lesion and resectability or oncologic outcomes.

## Figures and Tables

**Figure 1 cancers-13-03901-f001:**
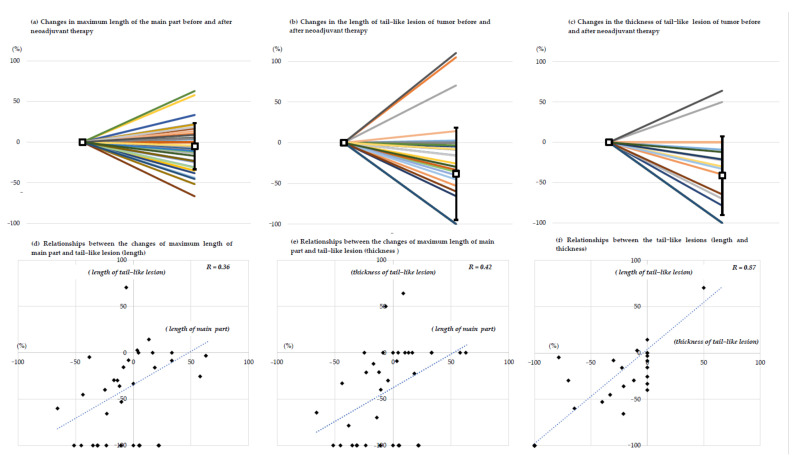
The rates of change in the tumor and the relationship between the main part and tail-like lesion. (**a**) Rates of change in maximum length of the tumor. (**b**) Rates of change in the length of the tail-like lesion. (**c**) Rates of change in the thickness of the tail-like lesion. (**d**) Relationship between the changes in maximum length of the main part and the length of the tail-like lesion. Squares indicate the means, and the error bars indicate the standard deviations. (**e**) Relationship between the rate of change in maximum length of the main part and the thickness of the tail-like lesion. (**f**) Relationship between the tail-like lesions’ length and thickness.

**Figure 2 cancers-13-03901-f002:**
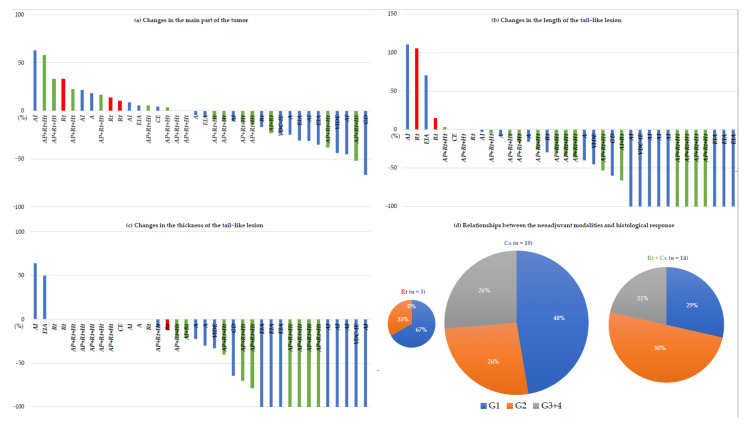
Waterfall plots illustrating the rates of change in the tumor with various neoadjuvant therapies. The rates of change were evaluated using the aforementioned method. (**a**) The rates of change in the main part of the tumor; (**b**) the rates of change in the length of the tail-like lesion; (**c**) the rates of change in the thickness of the tail-like lesion. The red bars indicate radiotherapy, blue bars indicate chemotherapy, and green bars indicate combination therapy (chemoradiotherapy); (**d**) the pie charts indicate the relationships between the neoadjuvant modalities and histological response (4-tier classification). For readability, grades 3 and 4 were integrated. In the Rt + Cx group, hyperthermia was added as an augmentation. A, Adriamycin; C, carboplatin; Cx, chemotherapy; D, docetaxel; E, etoposide; G, gemcitabine; Ht, hyperthermia; I, ifosfamide; P, cisplatin; Rt, radiotherapy; VDC/IE, vincristine + Adriamycin + cyclophosphamide/ifosfamide + etoposide; VIDE, vincristine + ifosfamide + Adriamycin + etoposide.

**Figure 3 cancers-13-03901-f003:**
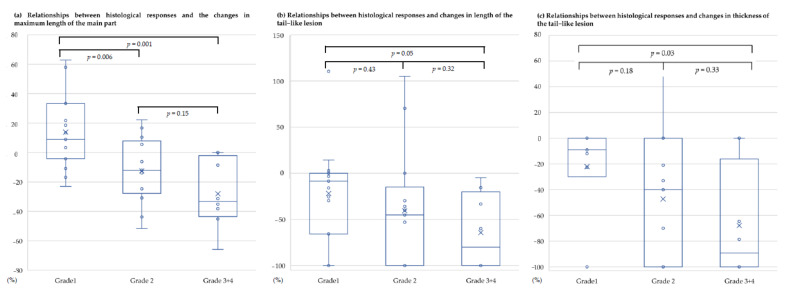
Box-and-whisker plots of the relationship between the histopathological evaluations and the shrinkage of tumors. (**a**) Relationships between the histopathological response and changes in the maximum length of the main part; (**b**) relationships between the histopathological response and changes in the length of the tail-like lesion; (**c**) relationships between the histopathological response and changes in the thickness of the tail-like lesion.

**Figure 4 cancers-13-03901-f004:**
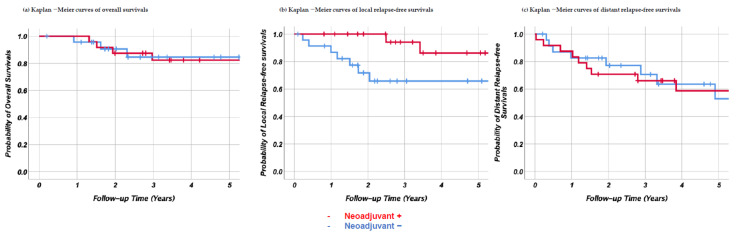
Kaplan–Meier curves comparing patients who underwent neoadjuvant therapy with those who did not, among UPS + MFS subtypes. (**a**) Overall survival, (**b**) local relapse-free survival, and (**c**) distant relapse-free survival. The red and blue lines indicate patients with and without neoadjuvant therapy, respectively.

**Table 1 cancers-13-03901-t001:** Characteristics of patients with a tumor with tail-like lesions.

Characteristics (*n* = 36)	Sub-Items	Number
Histology	UPS	11
	MFS	13
	SS	4
	DDL	4
	Others	4
Age at diagnosis (years, mean, standard deviation)	57.9, 15.5	
Tumor length (mm, median, IQR)	76.0, 53.3–93.3	
Tail-like lesion’s length (mm, median, IQR)	29.5, 23.0–37.3	
Tail-like lesion’s thickness (mm, median, IQR)	4.0, 2.2–7.7	
Location	Lower extremity	30
	Buttock	2
	Inguinal region	3
	Thigh	15
	Knee	3
	Lower leg	6
	Foot	1
	Upper extremity	3
	Upper arm	1
	Forearm	2
	Trunk	3
	Chest wall	1
	Back	2
Sex	Male	21
	Female	15
Lesion status	Primary	33
	Recurrence	3
FNCLCC grade	Grade 2	4
	Grade 3	32
Biopsy method	Needle	14
	Open	22
Depth	Superficial	13
	Deep	23

UPS, undifferentiated pleomorphic sarcoma; MFS, myxofibrosarcoma; SS, synovial sarcoma; DDL, dedifferentiated liposarcoma; IQR, interquartile range; FNCLCC, French Federation of Cancer Centers.

**Table 2 cancers-13-03901-t002:** Relationship between the histological response and changes in the main lesion (based on RECIST 1.1).

Characteristics		RECIST 1.1			Total
		PD	SD	PR	
Histological response	G1	5	10	0	15
	G2	1	9	3	13
	G3	0	3	4	7
	G4	0	0	1	1
Total		6	22	8	36

RECIST, Response Evaluation Criteria in Solid Tumors; PD, progressive disease; SD, stable disease; PR, partial response.

## Data Availability

The data presented in this study are available on request from the corresponding author. The data are not publicly available.

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
