# Peer review of "Effect of Neoadjuvant Therapies on Soft Tissue Sarcomas with Tail-like Lesions: A Multicenter Retrospective Study"

_cancers, 2021, doi:10.3390/cancers13153901_

Round 1
Reviewer 1 Report
I commend the authors on conducting an important study. Overall these findings, taken at face value with the limitations already raised by the author, are of interest. The authors performed a retrospective review of 31 cases of STS before and after neoadjuvant therapy. They performed thorough assessments with pathologists, and independent radiologists on the tumor specimens and radiographic data, respectively. The data are clearly presented. After neoadjuvant therapy, tail-like lesion’s shrinkage was observed in a significant proportion of patients. Although the authors rightfully concluded that they were unable to detect a relationship between tail-length and outcome, interesting findings were reported on neoadjuvant reception and tumor/disease features.
Minor suggestions:
Cannot see the legend on the waterfall plot.
More details are needed in the introduction to orient the reader - current data supporting/not supporting neoadjuvant therapy, treatment of early stage STS, heterogenity of disease
A brief consort diagram of included/excluded patients needed
Would be helpful to split the baseline characteristics table into 1) all patients 2) neoadjuvant recieved +/- not recieved to see if there were any differences between groups
Author Response
Responses to Comments of Reviewer 1
I commend the authors on conducting an important study. Overall these findings, taken at face value with the limitations already raised by the author, are of interest. The authors performed a retrospective review of 31 cases of STS before and after neoadjuvant therapy. They performed thorough assessments with pathologists, and independent radiologists on the tumor specimens and radiographic data, respectively. The data are clearly presented. After neoadjuvant therapy, tail-like lesion’s shrinkage was observed in a significant proportion of patients. Although the authors rightfully
concluded that they were unable to detect a relationship between tail-length and outcome, interesting findings were reported on neoadjuvant reception and tumor/disease features.
Minor suggestions:
Point 1: Cannot see the legend on the waterfall plot.
Response 1: We apologize for the incomprehensiveness of the legend. We have revised the legend of Figure 2 to:(Lines 224–232)
Figure 2. Waterfall plots illustrating the rates of change in the tumor with various neoadjuvant therapies. The rates of change were evaluated using the aforementioned method. (a) the rates of change in main part of the tumor; (b) the rates of change in the tail-like lesion (length); (c) the rates of change in the tail-like lesion (thickness). The red bars indicate radiotherapy, blue bars indicate chemotherapy, and green bars indicate combination therapy (chemoradiotherapy); (d) the pie graphs indicate the relationships between the neoadjuvant modalities and histological response (4-tier classification). For readability, grade 3 and 4 was integrated. In the Rt + Cx group, hyperthermia was added as an augmentation. A, Adriamycin; C, carboplatin; Cx, chemotherapy, D, docetaxel; E, etoposide; G, gemcitabine; Ht, hy-perthermia; I, ifosfamide; P, cisplatin; Rt, radiotherapy; VDC/IE, vincristine + Adriamycin + cyclophospha-mide/ifosfamide + etoposide; VIDE, vincristine + ifosfamide + Adriamycin + etoposide.
Point 2: More details are needed in the introduction to orient the reader - current data supporting/not supporting neoadjuvant therapy, treatment of early stage STS, heterogenity of disease
Response 2: Thank you for the suggestion. We have added data supporting and contradicting neoadjuvant therapy in the introduction section.
(Lines 65–69) Due to heterogeneity, the contribution of chemotherapy in the improvement of oncologic outcomes of soft tissue sarcomas were considered to be limited. If limited to high-risk cases (high grade malignancy grade, ≥5 cm in diameter, and deeply located with respect to investing fascia), the efficacy of chemotherapy was indicated.
Point 3: A brief consort diagram of included/excluded patients needed
Response 3: Thank you for the suggestion. We have added the description about the inclusion/exclusion of patients.
(Line 75-86) We included patients with histologically diagnosed malignant soft tissue tumors with tail-like lesions who underwent neoadjuvant therapy for primary soft tissue tumors between January 2012 and December 2019. Certified pathologists confirmed all diagno-ses at each hospital.We excluded patients with visceral location, metastasis (distant, skip lesion from the primary site, or lymph node metastasis) at diagnosis, and lack of images for proper evaluation. Further, we excluded patients who underwent amputation. Using independent questionnaires assigned to the 12 hospitals of the Tokai Musculoskeletal Oncology Consortium, 105 patients who underwent neoadjuvant therapy were extracted among a total of 951 patients. Among them, we finally included 36 patients who demon-strated tail-like lesion. The questionnaire included sex, age at diagnosis, histological di-agnosis, histological grade according to the French Federation of Cancer Centers [12], tumor location and depth, and (neo)adjuvant therapy details.
Point 4: Would be helpful to split the baseline characteristics table into 1) all patients 2) neoadjuvant recieved +/- not recieved to see if there were any differences between groups
Response 4: Thank you for the suggestion. The corresponding description has been depicted in Table 1 and Table A5 (supplementary file).

Reviewer 2 Report
The authors performed a retrospective study to asses the impact of neoadjuvant therapy on soft tissue sarcomas with a tail like lesion. This was a small study of 26 patients treated over 7 years. In addition it was a heterogenous group of tumors treated with heterogenous neoadjuvant therapy. One major flaw is that radiation was omitted in a handful of patients and based on the description of these tumor radiation would be standard of care. In general it is difficult to glean much from this study given heterogeneity of tumors and therapy and small numbers. There are many attempts in the paper to find associations of response in the tail to different factors. Honestly, I think what is most important about this paper is that local control is on par with what we would expect from all high grade soft tissue sarcomas (~90%). As such I believe the authors should emphasize that the tail should be treated with neoadjuvant therapy (radiation +/- chemotherapy) and from this small study these patient still have great local control.
Some other points:
Introduction:
-Would re-word paragraph 2 to explain the neoadjuvant radiation is standard of care for all stage II/III sarcomas regarless of functional outcome. Chemotherapy or hypothermia do not replace the need for radiation. I am also unaware of good data using hypothermia as neoadjuvant therapy.
Section 2.4- Why is response only to chemotherapy. What about radiation?
Table 1- clarify units for size. In paper this is mm, but would just also add in table for clarity.
Section 3.2-
I would be careful about categorizing patients as PD after neoadjuvant radiation. We often see some tumor swelling, but still with significant necrosis at time of surgery. What was the pathologic response rate of these tumors
This sentence is confusing “There was a partial statistical significance during this treatment (length, p = 0.088; thickness, p < 0.001) “ I would just state there was a statistical significant reduction in thickness (p<0.001), but not length (p=0.088)
Section 3.5- This is confusing what is the cohort who did not receive neoadjuvant therapy? Did they receive adjuvant therapy?
Discussion:
Neoadjuvant radiation is not controversial for infiltrative soft tissue sarcoma. Rather it is the standard of care regardless on impact on the tail.
Author Response
Response to Reviewer 2 Comments
Point 1: The authors performed a retrospective study to asses the impact of neoadjuvant therapy on soft tissue sarcomas with a tail like lesion. This was a small study of 26 patients treated over 7 years. In addition it was a heterogenous group of tumors treated with heterogenous neoadjuvant therapy. One major flaw is that radiation was omitted in a handful of patients and based on the description of these tumor radiation would be standard of care. In general it is difficult to glean much from this study given heterogeneity of tumors and therapy and small numbers. There are many attempts in the paper to find associations of response in the tail to different factors. Honestly, I think what is most important about this paper is that local control is on par with what we would expect from all high grade soft tissue sarcomas (~90%). As such I believe the authors should emphasize that the tail should be treated with neoadjuvant therapy (radiation +/- chemotherapy) and from this small study these patient still have great local control.
Response 1: I totally agree with your comment and emphasize that tail-like lesions should be treated with neoadjuvant therapy (radiation +/- chemotherapy). As the data we analyzed were from different centers, treatment strategy of tumor was versatile. Although this might be considered a major flaw, Japanese Orthoapedic Association (JOA) clinical practice guidelines on the management of soft tissue tumor had not recommended routine radiotherapy and, if radiotherapy is performed, preoperative or postoperative treatments are considered to be equal (O’Sullivan B, et al. Lancet 2002; 359: 2235-2241). However, some clinicians perform radiotherapy solely based on surgical difficulty (rough exposure of AVN) or microscopical residue cases.
Chemoradiotherapy should be emphasized, but it is difficult due to the lack of statistical backing. Nevertheless, we added the Figure 2d to support the efficacy of chemoradiotherapy and added the text in the discussion section.
(Lines 348–351)
Despite no statistical backing, we showed favorable results on the shrinkage of the tail lesion or in histopathological necrosis grade in patients treated with chemoradio-therapy. In case of resistance to radiotherapy or chemotherapy by the soft tissue sarcoma, these multimodal agents might be considered.
(Lines 377–383)
However, in some institution in Japan, the physicians do not perform routine radiotherapy because the Japanese Orthoapedic Association clinical practice guidelines on the management of soft tissue tumor does not recommend routine radiotherapy [37], despite decent evidence in support of radiotherapy [7, 38]. Some clinicians in Japan prefer to perform perioperative chemotherapy based on the decent result of clinical studies (JCOG0304, JCOG1306) performed in our nation. The difference among the modalities should be validated in a future study.
Point 2: Introduction:
-Would re-word paragraph 2 to explain the neoadjuvant radiation is standard of care for all stage II/III sarcomas regarless of functional outcome. Chemotherapy or hypothermia do not replace the need for radiation. I am also unaware of good data using hypothermia as neoadjuvant therapy.
Response 2:
Thank you for your comment on the introduction section. We totally agree with your comment, but this phrase was referred from the NCCN guideline 2021 (https://www.nccn.org/professionals/physician_gls/pdf/sarcoma.pdf). Based on your valuable comment, to avoid any misunderstanding, we have slightly changed the description.
(Lines 64–70)
These methods include radiotherapy [7], chemoradiotherapy [8], or chemotherapy [9,10], and hyperthermia (e.g., isolated limb perfusion therapy) [11]. Due to heterogeneity, the contribution of chemotherapy in the improvement of oncologic outcomes of soft tissue sarcomas was considered to be limited. However, if limited to high-risk cases (high grade malignancy, ≥5 cm in diameter, and deeply located with respect to investing fascia), the efficacy of the chemotherapy has been indicated.
Point 3: Section 2.4- Why is response only to chemotherapy. What about radiation?
Response 3:
We apologize for not adding it. We understand the description was inadequate. We have changed it.
(Lines 137–141)
• Grade 1: little or no effect of neoadjuvant therapy observed
• Grade 2: partial response to neoadjuvant therapy with >50% tumor necrosis
• Grade 3: >90% tumor necrosis attributable to preoperative neoadjuvant therapy,
Point 4: Table 1- clarify units for size. In paper this is mm, but would just also add in table for clarity.
Response 4:
We apologize for the mistake. We have added the units in the tables.
Point 5: Section 3.2- I would be careful about categorizing patients as PD after neoadjuvant radiation. We often see some tumor swelling, but still with significant necrosis at time of surgery. What was the pathologic response rate of these tumors
Response 5:
Thank you for the query. We totally agree with your concerns. There was a case with rapid increase of tumor in size (categorized as PD in RECIST 1.1) with attenuated the contrast agent and internal necrosis in the central part of tumor. As you said, evaluating the response to neoadjuvant therapy, including radiotherapy is difficult based only on the size of the tumor. RECIST 1.1 is ubiquitously used as the tool for the evaluation of the response to some agents; however, it is based only on the size. Recently, modified RECIST criteria (based on the accumulation of contrast agents on MRI), choi criteria, MD Andarson criteria, PERCIST and so on, were introduced as cancer response criteria. However, as the present study was retrospective in nature, we integrated all the acceptable data via RECIST 1.1 criteria as an evaluation tool. Various key studies also used RECIST 1.1 criteria for evaluation of neoadjuvant therapy (EORTC-62092: STRASS study, ISG-STS 1001). We have added the description about inadequate evaluation methods under limitations in the discussion section. A future study should evaluate the efficacy of neoadjuvant therapy using more reliable methods (Line 383-389).
Moreover, we have added the data about the relationship between the histological response, deemed actual effect of neoadjuvant therapy, and the versatile neoadjuvant methods in the Figure 2d.
Point 6: This sentence is confusing “There was a partial statistical significance during this treatment (length, p = 0.088; thickness, p < 0.001) “ I would just state there was a statistical significant reduction in thickness (p<0.001), but not length (p=0.088)
Response 6: Thank you for pointing this out. We have changed the description regarding this.
(Lines 199–200)
There was a statistically significant reduction in thickness (p<0.001), but not in length (p=0.088).
Point 7: Section 3.5- This is confusing what is the cohort who did not receive neoadjuvant therapy? Did they receive adjuvant therapy?
Response 7: Thank you for the query. This was based on the physician’s choice, but the major reasons why these patients did not receive neoadjuvant therapy were small size, depth of tumor, intolerance to the chemotherapy due to age, and resectability of the tumor. Supplemental table 5a demonstrates these differences. Moreover, we have added the information about post-operative radiotherapy.
Point 8: Discussion:
Neoadjuvant radiation is not controversial for infiltrative soft tissue sarcoma. Rather it is the standard of care regardless on impact on the tail.
Response 8: We apologize for the confusing sentence. We know that the radiotherapy is effective in infiltrative soft tissue tumor. Rather, we wanted to convey the efficacy of neoadjuvant therapy on tail-like lesions. We have revised the sentences in the manuscript to:
(Lines 338–342)
The effect of neoadjuvant therapy on the tail-like lesion remains controversial. Several studies concluded that preoperative radiotherapy had no effect on the tail sign [26], while others reported positive results [31,32]. These conflicting viewpoints were due to the differences in sample size of the studies or the method used to evaluate the efficacy of neo-adjuvant therapy on tail-like lesion.
